# Effectiveness of Preoperative Immunonutrition in Improving Surgical Outcomes after Radical Cystectomy for Bladder Cancer: Study Protocol for a Multicentre, Open-Label, Randomised Trial (INu-RC)

**DOI:** 10.3390/healthcare12060696

**Published:** 2024-03-20

**Authors:** Valentina Da Prat, Lucia Aretano, Marco Moschini, Arianna Bettiga, Silvia Crotti, Francesca De Simeis, Emanuele Cereda, Amanda Casirati, Andrea Pontara, Federica Invernizzi, Catherine Klersy, Giulia Gambini, Valeria Musella, Carlo Marchetti, Alberto Briganti, Paolo Cotogni, Richard Naspro, Francesco Montorsi, Riccardo Caccialanza

**Affiliations:** 1Clinical Nutrition and Dietetics Unit, Fondazione IRCCS Policlinico San Matteo, 27100 Pavia, Italyr.caccialanza@smatteo.pv.it (R.C.); 2Department of Urology, Fondazione IRCCS Policlinico San Matteo, 27100 Pavia, Italy; 3Department of Urology, IRCCS Ospedale San Raffaele, 20132 Milan, Italy; 4Clinical Nutrition, IRCCS Ospedale San Raffaele, 20132 Milan, Italy; 5Division of Internal Medicine and Hepatology, Center for Liver Disease, IRCCS Ospedale San Raffaele, 20132 Milan, Italy; 6Clinical Epidemiology and Biometry Service, Fondazione IRCCS Policlinico San Matteo, 27100 Pavia, Italy; 7Pain Management and Palliative Care, Department of Anesthesia, Intensive Care and Emergency, Molinette Hospital, University of Turin, 10126 Turin, Italy

**Keywords:** immunonutrition, radical cystectomy, postoperative complications, bladder cancer

## Abstract

Radical cystectomy (RC) with pelvic lymph node dissection is the standard treatment for patients with limited-stage muscle-invasive bladder cancer. RC is associated with a complication rate of approximately 50–88%. Immunonutrition (IMN) refers to the administration of substrates, such as omega-3 fatty acids, arginine, glutamine, and nucleotides, that modulate the immune response. IMN has been associated with improved outcomes following surgery for esophagogastric, colorectal and pancreatic cancer. In this paper, we describe a study protocol for a multicentre, randomised, open-label clinical trial to evaluate the effect of IMN in patients undergoing RC for bladder cancer. A 7-day preoperative course of IMN is compared with a standard high-calorie high-protein oral nutritional supplement. The primary outcome of this study is the rate of complications (infectious, wound-related, gastrointestinal, and urinary complications) in the first 30 days after RC. Secondary outcomes include time to recovery of bowel function and postoperative mobilisation, changes in muscle strength and body weight, biochemical modifications, need for blood transfusion, length of stay, readmission rate, and mortality. The results of this study may provide new insights into the impact of IMN on postoperative outcomes after RC and may help improve IMN prescribing based on patient nutritional status parameters.

## 1. Introduction

Bladder cancer (BC) is the 10th most common cancer type worldwide, with the highest incidence rates observed in Greece, the Netherlands, Italy, Denmark, and Belgium [1]. The main risk factor for developing BC is tobacco smoking, which accounts for up to 50% of cases, followed by occupational exposure to aromatic amines and ionising radiation [2]. The majority of patients with bladder cancer are diagnosed with non-muscle-invasive disease (NMIBC; 75% of cases), which can be treated with transurethral resection of the bladder tumour, intravesical instillation of Bacillus Calmette-Guérin (BCG), or radical cystectomy (RC) with pelvic lymph node dissection (PLND) in very high-risk and BCG-unresponsive cases. Muscle-invasive BC (MIBC) accounts for 25% of cases of BC. RC with PLND is the standard of care for limited-stage MIBC. Patients with clinical node-positive disease may benefit from neoadjuvant platinum-based chemotherapy before surgery. RC is followed by urinary diversion and reconstruction. The latter usually involves resection of an isolated bowel segment (usually the distal ileum) for continent orthotopic (neobladder) or incontinent cutaneous (ileal conduit, cutaneous uretherostomy) reconstruction, depending on the patient’s clinical condition. In particular, neobladder reconstruction can be offered to patients who have no contraindication and no tumour at the urethral level [3].

RC with PLND and urinary reconstruction (hereafter referred to as RC) is associated with an overall complication rate of around 64% [4]. In particular, postoperative complications are usually graded according to the Clavien–Dindo classification [5]. The latest European guidelines describe an incidence of all-grade complications of 50–88% (Clavien–Dindo grade I–IV) and severe complications of 30–42% (Clavien–Dindo grade ≥ III) [2]. Gastrointestinal complications are the most common (29%), including postoperative ileus in up to 26% of patients. Infectious complications (25%) and wound-related complications (15%) are also very common [4].

Preoperative nutritional status is a critical aspect to consider in patients undergoing RC. In these patients, malnutrition may be caused by the combined action of cancer disease, ageing, and the effects of neoadjuvant chemotherapy, which adversely affect oral intake. Accordingly, the estimated prevalence of malnutrition in the RC population ranges from 21% to 55% using the Nutritional Risk Screening 2002 (NRS-2002) tool [6]. Notably, nutritional status prior to RC has been shown to predict 90-day mortality and poor overall survival [7]. In addition, malnutrition is often associated with sarcopenia, which can lead to an increased risk of hospital readmission, longer hospital stays, and higher mortality rates [8]. While conclusive data on the impact of sarcopenia in MIBC patients are lacking due to limited evidence, skeletal muscle loss appears to be a significant predictor of 90-day mortality and postoperative complications after RC [9].

Nutritional support should ideally be initiated when patients are not yet malnourished to prevent the development of malnutrition and sarcopenia; therefore, the early assessment of nutritional risk is mandatory. The first type of nutritional support should be dietary counselling, with or without the administration of oral nutritional supplements (ONSs). If oral intake is impossible or insufficient, artificial nutrition should be started. If possible, enteral nutrition should always be preferred to parenteral nutrition, as it has a more favourable safety profile due to reduced infectious complications and preservation of the gut microbiota. In clinical practice, RC is often followed by a long period of fasting, despite the recommendations of Enhanced Recovery After Surgery (ERAS). Total parenteral nutrition (TPN) is widely used to counteract the protein catabolism that occurs in the early phases after RC [10].

The term “immunonutrition” (IMN) refers to the oral or enteral administration of specific substrates, such as arginine, glutamine, omega-3 fatty acids, and nucleotides. These substrates have been shown to upregulate the host immune response, modulate inflammatory responses, and improve protein synthesis after surgery. In particular, arginine is a “conditionally essential” amino acid (i.e., its synthesis may be limited under certain pathophysiological conditions), and its levels decrease rapidly during postoperative stress. Arginine deficiency can reduce nitric oxide, collagen, T-cell function, and protein translation, leading to thrombosis, increased susceptibility to infection, wound breakdown, and muscle wasting [11,12]. Increased interleukin-2 (IL-2) production and receptor activity have been demonstrated in arginine-supplemented animals, while a favourable impact on survival has been shown in animal models of peritonitis [13]. Glutamine is an important fuel for the intestine and lymphocyte proliferation and is also a precursor of glutathione. Animal studies have shown preserved or improved intestinal integrity with reduced bacterial translocation in glutamine-supplemented animals [13]. The importance of omega-3 fatty acids is related to their metabolites, eicosapentanoic acid and docosahexanoic acid, which are associated with a reduced suppression of cell-mediated immune responses, lymphokine production, and cell-mediated cytolysis, as shown in animal studies [13]. Finally, nucleotides appear to be essential for maintaining immune function. Studies in rodents have shown that a nucleotide-free diet results in impaired T-cell-mediated immunity; moreover, mice fed either with a protein-free or a nucleotide-free diet required either RNA or uracil in the diet in order to reverse the immunosuppression, suggesting that neither calories nor protein alone are sufficient to reverse the immune effects of protein deprivation [14].

Because major surgery, like other types of injury, is associated with a dysfunction of host homeostasis, immune mechanisms, and inflammatory response, the effects of IMN have been extensively studied in clinical trials in the surgical setting. In the last three decades, perioperative administration of IMN has been shown to reduce both postoperative infection rates and length of hospital stay for major abdominal surgery for oesophageal, gastric, colorectal, and pancreatic cancer [15]. The latest European Society of Clinical Nutrition and Metabolism (ESPEN) guidelines recommend the provision of IMN perioperatively or at least postoperatively for malnourished patients undergoing major abdominal cancer surgery [16].

In the light of studies performed in the gastrointestinal surgical setting, it is reasonable to expect a beneficial effect of IMN in patients undergoing RC, which is a major intraperitoneal surgical procedure that often includes bowel resection for urinary reconstruction. However, data on IMN in patients undergoing RC are scarce, mainly due to the small sample size of the trials and the high heterogeneity between studies [17]. Several aspects remain largely unclear, including the dosage of IMN, the timing of administration, and the impact of preoperative malnutrition risk on IMN efficacy.

## 2. Materials and Methods

### 2.1. Study Design and Setting

This is a controlled, randomised, open-label, parallel-group clinical trial to evaluate the effect of preoperative IMN on postoperative outcomes after RC for BC.

A 7-day preoperative treatment with IMN (cases) will be compared with a 7-day preoperative treatment with standard high-calorie high-protein ONS (controls).

Informed consent will be obtained from each patient. Patients will have the right to withdraw their consent at any time without altering their current treatment.

This study will be conducted in two tertiary hospitals: (1) the Fondazione IRCCS Policlinico San Matteo, Pavia, Italy; (2) the IRCCS Ospedale San Raffaele, Milan, Italy.

### 2.2. Participants

Participants who meet all of the following inclusion criteria are eligible for the study: signed informed consent; surgical indication to RC with a diagnosis of MIBC (any N, any M), BCG-unresponsive NMIBC, or extensive NMIBC that cannot be treated with endoscopic surgery alone.

The presence of any one of the following exclusion criteria will lead to the exclusion of the participant: age < 18 years; pregnancy or breastfeeding; contraindications to the nutritional supplements under study, e.g., known hypersensitivity or allergy to the supplements; need for artificial nutrition due to total impairment of spontaneous food intake; diarrhoea with suspected malabsorption syndrome; participation in another study with nutritional supplements within 30 days before and during the present study; inability to take oral supplements due to a pre-existing condition (e.g., dysphagia) or other factors (e.g., language barrier, psychological disorders, lack of home care in dependent or elderly patients); renal failure requiring renal replacement therapy; type 1 diabetes mellitus; type 2 diabetes mellitus requiring insulin therapy and/or with inadequate glycaemic control (glycosylated haemoglobin ≥ 7% and/or fasting plasma glucose ≥ 150 mg/dL).

### 2.3. Recruitment and Randomization of Study Participants

Participants will be recruited during the pre-admission dietary visit, 2–3 weeks prior to surgery. The screening procedure will only require routine/daily practice examinations to check that the patient meets the inclusion and exclusion criteria. In particular, screening for inclusion and exclusion criteria will be based on the patient’s clinical history, home medication, routine pre-admission laboratory tests, dietary intake, and presence/absence of caregivers in the case of dependent or elderly patients. No payment or compensation will be provided to study participants. After signing the informed consent form, eligible participants will be randomised 1:1 by the treating physician to one of the two study arms according to a computer-generated random block randomisation list. Randomisation will be stratified by centre.

### 2.4. Nutritional Interventions

The study intervention consists of the preoperative administration of a high-calorie, high-protein, immunonutrient-enriched ONS (Impact^®^ Oral (237 mL per dose); Nestlé Health Science—Creully Sur Seulles—France). Two doses of Impact^®^ Oral per day will be given from 7 days before surgery until the day before RC. Specifically, this formula is enriched in immunonutrients (arginine, RNA nucleotides and omega-3 fatty acids) and provides 36 g of protein and 682 kcal/day at the proposed therapeutic dose.

Patients in the control group will receive a high-calorie, high-protein liquid ONS (Meritene^®^ Protein Drink (200 mL per serving); Nestlé Health Science—Creully Sur Seulles—France). Two doses of Meritene^®^ Protein Drink per day will be given from 7 days before surgery until the day before RC. Specifically, this formula provides 38 g of protein and 500 kcal/day at the proposed therapeutic volume.

In the event of unplanned delays in the surgical procedure, the 7-day administration of IMN/ONS will be temporarily interrupted and then resumed, with the last dose taken the day before surgery (e.g., if surgery is delayed after a patient has already taken 5 doses of IMN/ONS, he/she will take the remaining 9 doses in the 5 days before the new surgery date).

To improve compliance, nutritional counselling will be provided at the pre-admission visit in both treatment arms. Patients will be advised on how to take the supplements (i.e., they will be instructed to drink the supplements very slowly between meals) and will be given an information leaflet on how to take the supplements. Overweight and obese patients will be given dietary advice on how to reduce caloric intake to avoid an excessive increase in daily caloric intake. Patients will be asked to return any unused product on admission to the hospital in order to quantify the amount of product consumed.

Patients from both participating centres will follow the same pre- and postoperative routines. On the day before surgery, dinner will consist of a light liquid meal (e.g., broth). In accordance with the principles of Enhanced Recovery After Surgery (ERAS), all patients will be given an iso-osmolar carbohydrate oral supplement, with sugars and sweeteners and without proteins, fats, and fibre, up to two hours before surgery. After surgery, oral feeding is resumed when bowel function has recovered. An appropriate postoperative diet will be used to gradually increase caloric intake and food complexity. If nutritional intake is inadequate due to reduced appetite, dyspepsia, or other symptoms, standard ONS can be used in both groups to meet caloric and nutrient requirements.

If early oral refeeding is not feasible, patients with a low preoperative risk of malnutrition (NRS-2002 < 3) will receive intravenous hydration until postoperative day 4. At this time, if oral feeding is not possible and there are no contraindications, TPN will be prescribed until the patient is able to resume oral feeding. In patients at high risk of preoperative malnutrition (NRS-2002 ≥ 3), TPN will be started from postoperative day 1–2 unless contraindicated. The post-operative algorithm for TPN usage is detailed in Appendix A. Variations in the timing and composition of TPN are allowed at the clinician’s discretion.

### 2.5. Outcomes and Objectives

The aim of this study is to evaluate the impact of preoperative IMN compared to standard high-calorie high-protein ONS on surgical outcomes in patients undergoing RC for BC.

The primary objective of this study is to compare preoperative IMN with standard high-calorie-high-protein ONS in terms of the rate of complications (infectious, wound-related, gastrointestinal, and urinary complications) in the first 30 days after RC for BC.

The key secondary objective is to compare preoperative IMN with standard ONS in terms of the rate of severe complications (i.e., Clavien–Dindo grade ≥ III) in the first 30 days after RC. The other secondary objectives of the study are as follows:To compare preoperative IMN with standard ONS in terms of the rate of severe complications in the first 90 days after RC;To compare preoperative IMN with standard ONS in terms of the rate of infectious complications in the first 30 and 90 days after RC;To compare preoperative IMN with standard ONS in terms of the rate of occurrence of other medical conditions in the first 30 and 90 days after RC;To compare preoperative IMN with standard ONS in terms of time to recovery of bowel function, defined as the time from surgery to first flatus;To compare preoperative IMN with standard ONS in terms of time to postoperative mobilization, defined as the time from surgery to first walk;To compare preoperative IMN with standard ONS in terms of changes in muscle strength, defined as the difference in hand-grip strength (mean of three consecutive measurements from the dominant hand) and in the sit-and-stand test from the preadmission visit to the day before surgery to hospital discharge;To compare preoperative IMN with standard ONS in terms of weight modifications, defined as the difference in body weight from the preadmission visit to the day before surgery to hospital discharge to the 90-day follow-up visit;To compare preoperative IMN with standard ONS in terms of biochemical nutritional index modifications, i.e., changes in haemoglobin, lymphocytes, albumin, prealbumin, C-reactive protein, cholinesterase, total cholesterol, glucose, and creatinine, from the preadmission visit to the day before surgery to hospital discharge;To compare preoperative IMN with standard ONS in terms of the need for blood transfusions, defined as the percentage of patients who receive blood transfusions during their hospital stay;To compare preoperative IMN with standard ONS in terms of length of stay (LOS), defined as the time from hospital admission to discharge at home or to another facility;To compare preoperative IMN with standard ONS in terms of 30-day and 90-day readmission rate, defined as the incidence of unplanned re-hospitalization due to all causes in the first 30 and 90 days;To compare preoperative IMN with standard ONS in terms of 30-day and 90-day mortality, defined as death rate due to all causes during the first 30 and 90 days post-surgery;To compare the compliance to preoperative IMN and standard ONS, defined as the percentage of patients consuming ≥ 80% of the prescribed supplement in the 7 days before surgery;To compare the tolerability of preoperative IMN and standard ONS, defined as the percentage of patients experiencing at least one moderate–severe adverse gastrointestinal effects during the 7-day intervention period;To compare preoperative IMN with standard ONS for the primary endpoint and the key secondary endpoint in the subgroups of the following:
Patients with preoperative diagnosis of type II diabetes vs. non-diabetic patients;Overweight/obese patients (i.e., patients with preadmission BMI ≥ 25/30, respectively) vs. non-overweight/obese patients;Patients aged ≥ 70 vs. <70;Patients with NRS-2002 ≥ 3 vs. <3.


Complications and other medical conditions assessed are listed in Table 1.

Any participant may discontinue participation in the study because of the withdrawal of informed consent, a change in surgical indication (i.e., the patient is no longer undergoing RC), death, or the start of artificial nutrition.

### 2.6. Data Collection Methods

Data will be collected at five time points: during the screening visit, at hospital admission, at hospital discharge, at 30 days after surgery (by telephone), and 90 days after surgery.

Screening visit. The screening visit takes place during the pre-admission visit by the referring dietitian/nutritional biologist, usually 14–28 days before surgery. After evaluation of the inclusion and exclusion criteria and signing of informed consent, the following data will be collected: demographics; medical history and medications; laboratory data; pregnancy test in women of childbearing potential; anthropometric parameters; nutritional risk according to NRS-2002; body composition via analysis of basal CT scan images (if performed in the two months prior to the preadmission visit); muscular strength by digital hand-grip dynamometer and sit-and-stand test.Hospital admission. Hospital admission is usually the day before surgery. The following data will be recorded: anthropometric parameters; muscular strength as described above; compliance based on the number of servings of supplements consumed by the patient; tolerability measured using a questionnaire.Hospital discharge. Hospital discharge is usually 15 to 20 days after surgery. The following data will be recorded: anthropometric parameters; muscle strength as above; type of urinary reconstruction; day of recovery of bowel function; day of discontinuation of systemic/peridural opioid analgesic therapy, if any; day of removal of naso-gastric tube, if any; use of parenteral nutrition and/or ONS after surgery; blood transfusions performed during hospital stay; need for ICU stay; vital status.Thirty days after surgery. A follow-up telephone call will be made 30 days after surgery. The following data will be collected, if available: postoperative complications and other medical conditions; re-hospitalisation; vital status. If patients are still hospitalised 30 days after surgery, data will be collected during hospitalisation.Ninety days after surgery. A follow-up visit will be performed ninety days after surgery. The following data will be collected, if available: anthropometric parameters; postoperative complications and other medical conditions; re-hospitalisation; initiation of adjuvant chemotherapy; vital status. If patients are still hospitalised 90 days after surgery, data will be collected during hospitalisation.

### 2.7. Data Management and Statistical Analysis

#### 2.7.1. Determination of Sample Size

Sample size calculations are based on the primary endpoint. We estimate an expected absolute reduction of 17% in the complication rate or patients receiving preoperative IMN compared to patients receiving preoperative standard ONS. Postoperative complications following RC are known to occur in approximately 64–70% of patients without nutritional supplementation [4,18]. A small randomised pilot study showed a 30-day complication rate of around 73% in patients receiving standard ONS [19]. Data on patients receiving IMN are heterogeneous, due to high variability in IMN dosage and description of outcomes in the literature. The clinical studies evaluating the impact of IMN on patients undergoing RC report 30-day complication rates of 40% [20], 63% [21], 55.4% [22], and 71% [19]. In this study, we estimated an expected complication rate of 72% in the standard ONS group and 55% in the IMN group. The expected complication rate in the standard ONS group was derived from a previous study that enrolled only males and provided both pre- and postoperative standard ONS supplementation [19], while the estimation of the complication rate in the intervention group was derived from clinical studies on IMN [19,20,21,22].

Using the expected complication rates, a sample size calculation was performed (80% power, and a two-sided type I error = 0.05); a minimum 125-patient sample size was required in each group. The number of subjects was increased up to 130 in each group to take into account a potential drop-out rate of 4%. Therefore, the total number of patients enrolled for the entire study (all sites combined) will be 260. Based on the expected accrual rate, the study duration will take place over 30 months.

#### 2.7.2. Planned Analyses

The main analysis will be conducted according to the intention-to-treat (ITT) principle: patients in the analysis set will be analysed according to the treatment to which they were randomised, regardless of the treatment actually received. The ITT population will be used for the primary and secondary endpoints. Per-protocol (PP) analyses will also be performed, taking into account the treatment actually received. Specifically, per-protocol analyses will compare compliant patients in the IMN group with compliant patients in the control group for the primary and key secondary endpoints. The safety population will include all patients who consume at least one dose of the assigned treatment and will be used for the safety analyses. The two-sided type I error will be set at 5% (adjusting for Bonferroni’s correction when necessary). All statistical analyses will be performed using Stata 18 (StataCorp. 2021. Stata Statistical Software: Release 18. College Station, TX, USA: StataCorp LLC.)

The primary objective will be analysed using a binomial regression model, with link identity. The risk difference and its 95% confidence interval (95%CI) will be derived from the model. Huber–White robust standard errors will be used while clustering on centre to account for the lack of independence within centre.

The statistical analysis for the key secondary objective and for several other secondary objectives (no. 9, 13, and 14) will be performed in the same way as the primary objective. Other secondary objectives will be analysed using a time-to-event analysis (no. 1–5 and 10–12) or with Mann–Whitney U test (no. 6–8). Effect modification by subgroup will be assessed by introducing the interaction of subgroup and treatment in the corresponding statistical model (no. 15).

## 3. Discussion

Several studies have investigated the role of perioperative IMN in patients with bladder cancer undergoing surgery. Data are scarce, mainly due to the small sample sizes of these studies [23]. Moreover, some trials show controversial results. Nevertheless, the few available studies suggest a potential beneficial effect of IMN in this context [17].

Bertrand and colleagues were the first to evaluate the use of preoperative IMN (preoperative 7-day administration, 3 doses per day) in patients undergoing RC for BC with a retrospective case–control study; the authors observed a 48% relative reduction in the 30-day complication rate (40 vs. 76.7%) and lower rates of paralytic ileus at postoperative day 7 compared with patients who did not receive IMN [20]. Lyon and colleagues reported no improvement in the 90-day complication rate in patients treated with IMN (5-day preoperative administration, 4 doses per day) compared with no IMN [24]. Similarly, in a retrospective study, Cozzi and colleagues did not report lower rates of postoperative complications in RC candidates treated with perioperative IMN (7-day preoperative administration, 3 doses per day, and 7-day postoperative administration, 2 doses per day) [25]. However, Khaleel and colleagues retrospectively described a reduction in postoperative TPN requirements and infectious complication rates (25 vs. 45%) with IMN (5-day administration, 1 dose per day) [21]. The impact of IMN in the context of the ERAS protocol was recently evaluated in a case–control study comparing a historical cohort with IMN (5-day preoperative administration, 3 doses per day) in conjunction with preoperative carbohydrate load according to the ERAS protocol; no differences in complication and re-admission rates were detected [22].

Very few studies have compared IMN with standard ONS. Hamilton-Reeves and colleagues found that perioperative IMN versus standard ONS was associated with a 33% relative reduction in the postoperative complication rate and a 39% reduction in the infection rate during late-phase recovery, whereas no difference was found in the first 30 days (71% vs. 73%). In addition, lower levels of total myeloid-derived suppressor cells and plasma IL-6 were found on postoperative day 2 [19]. The same group reported a favourable shift in the Th1-Th2 balance from baseline to the intraoperative period and a trend towards less profound muscle loss in patients receiving IMN compared to standard ONS at postoperative day 14 [26]. A randomised phase III double-blind clinical trial is currently investigating the role of perioperative IMN with a novel supplement containing arginine, omega-3 fatty acids, nucleotides, and vitamin A in patients undergoing RC [27].

Of note, several aspects remain largely unclear, including the dosage and timing of IMN within enhanced recovery pathways and the impact of preoperative malnutrition risk on IMN efficacy.

## 4. Conclusions

Our study will help to clarify the impact of preoperative IMN in patients undergoing RC for BC. If successful, there is potential for preoperative IMN to become the standard of care for patients undergoing RC. In addition, the results of this study may provide new insights into the impact of malnutrition on postoperative outcomes and may help to improve IMN prescribing based on nutritional status parameters. Furthermore, our study on patients undergoing RC for BC may provide valuable information for investigating the role of preoperative IMN in other urinary surgeries, including nephrectomy or radical prostatectomy.

## Figures and Tables

**Table 1 healthcare-12-00696-t001:** Complications and other postoperative medical conditions evaluated in the study.

Complications	
Infectious complications	Abdominal abscess Bloodstream infection *C. difficile* colitis Cholecystitis Diverticulitis Pneumonia Pyelonephritis Wound infection with systemic signs
Gastrointestinal complications	Anastomotic bowel leak/fistula Functional ileus Gastrointestinal bleeding requiring transfusion Small bowel obstruction
Wound-related complications	Hernia or evisceration Wound dehiscence Wound infection without systemic signs
Genito-urinary complications	Urinary bleeding requiring transfusion Symptomatic lymphocele Ureteral obstruction Urinary fistula/leak
**Other Medical Conditions**	
Cardio-pulmonary complications	Congestive heart failure Myocardial infarction New onset of arrhythmia Pleural effusion requiring drainage
Thrombotic complications	Deep venous thrombosis Peripheral arterial ischemia Pulmonary embolism
Neurologic complications	Intracranial haemorrhage Delirium requiring psychiatric therapy Stroke Transient ischemic attack
Miscellaneous	Hyperglycaemia requiring insulin therapy in non-diabetic patients New onset of pressure ulcers Other

## Data Availability

No new data were created or analyzed in this study. Data sharing is not applicable to this article.

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
