# Peer review of "Effectiveness of Preoperative Immunonutrition in Improving Surgical Outcomes after Radical Cystectomy for Bladder Cancer: Study Protocol for a Multicentre, Open-Label, Randomised Trial (INu-RC)"

_healthcare, 2024, doi:10.3390/healthcare12060696_

Round 1

Reviewer 1 Report

Comments and Suggestions for Authors

An interesting study which should be premised on animal or other studies. Authors should build the scientific rationale for the study in the introduction.

Author Response

Thank you very much for taking the time to review this manuscript.

Comment 1. An interesting study which should be premised on animal or other studies. Authors should build the scientific rationale for the study in the introduction

Response 1. Thanks for pointing this out. We agree with your comment. We added a paragraph on animal studies and explained the rationale of using immunonutrition in patients undergoing radical cystectomy in the Introduction (page 3). Clinical studies on immunonutrition are listed in the Discussion section.

Reviewer 2 Report

Comments and Suggestions for Authors

The manuscript presents the protocol of a study by which the authors want to test the efficacy of a treatment with a immunonutrition course to patients undergoing radical cystectomy, comparing it with standard practices.

The Authors clearly explain the context in which they intend to intervene and the rationale for the trial. The sections of the manuscript address the different aspects of the trial comprehensively; however, I propose two questions to the Authors.

It is helpful if they could provide clarification on the expected duration of this trial (with the increasing proposals for treatments aimed at improving outcome in patients undergoing major surgery, the results of this trial could be anticipated by other new evidence). In addition, I think it is necessary for the authors to explain in more detail the estimated incidence of complications they used as an expected value in the untreated patient arm.

Author Response

We would like to thank the reviewer for raising these important issues and helping to improve this paper.

Comment 1. The manuscript presents the protocol of a study by which the authors want to test the efficacy of a treatment with a immunonutrition course to patients undergoing radical cystectomy, comparing it with standard practices. The Authors clearly explain the context in which they intend to intervene and the rationale for the trial. The sections of the manuscript address the different aspects of the trial comprehensively; however, I propose two questions to the Authors. It is helpful if they could provide clarification on the expected duration of this trial (with the increasing proposals for treatments aimed at improving outcome in patients undergoing major surgery, the results of this trial could be anticipated by other new evidence).

Response 1. We have added a sentence regarding the expected duration of the trial (2.5 years; page 8). Thank you for pointing this out. We agree that new knowledge may emerge during this period and lead to changes in clinical practice. Nevertheless, we believe that testing a relatively short preoperative course of immunonutrition could help to understand the best timing and dosage of immunonutrition supplements, especially given that other ongoing trials are evaluating perioperative (pre+post) administration.

Comment 2. In addition, I think it is necessary for the authors to explain in more detail the estimated incidence of complications they used as an expected value in the untreated patient arm.

Response 2. We clarified the choice of the expected complication rates used for sample size calculation (page 8). Thank you for highlighting this point.

Reviewer 3 Report

Comments and Suggestions for Authors

This is a very well written protocol paper and very well designed trial. The study was designed with a lot of details looking into the benefits of preoperative nutritional therapy, which may provide a better solution to minimize bladder cancer surgery complications.

Strength:

Writing English is perfect, no apparent grammatical errors.

Intro: provided enough back ground information and addressed the importance of nutritional therapy in bladder cancer surgery

Study design: very logical with good details. methods and data analysis also very well described with all the questions listed

Method is clearly described and analysis is well thought.

Weakness:

Not sure if the protocol is already been written, would be great to see the details in conducting the trial. 

The study topic is very important in practice, however this trial only included relatively healthy patients who mostly recover well from major surgeries given the exclusion criteria, which is understandable. It would be interesting to see future works on more critical population. 

In the manuscript the author stated IVF would be given if patients not able to start early oral intake. Pure fluid have no calories nor nutrients, not sure if this would impact the final results.

The solution for patients who might not be able to tolerate early oral nutrition s/p surgery not optimal. Would like to have a protocol attached as a supplement material

Author Response

Thank you very much for highlighting the strengths and weaknesses of this study.

Comment 1. This is a very well written protocol paper and very well designed trial. The study was designed with a lot of details looking into the benefits of preoperative nutritional therapy, which may provide a better solution to minimize bladder cancer surgery complications. Strength: Writing English is perfect, no apparent grammatical errors; Intro: provided enough back ground information and addressed the importance of nutritional therapy in bladder cancer surgery; Study design: very logical with good details. methods and data analysis also very well described with all the questions listed; Method is clearly described and analysis is well thought.

Response 1. Thank you for your comment.

Comment 2. Weakness: Not sure if the protocol is already been written, would be great to see the details in conducting the trial. 

Response 2. Thank you for your interest in this study. The protocol has been approved by the local ethics committee. We have tried to include as much detail as possible; we have uploaded some supplementary material (see comment 4) to further describe the nutritional interventions.

Comment 3. The study topic is very important in practice, however this trial only included relatively healthy patients who mostly recover well from major surgeries given the exclusion criteria, which is understandable. It would be interesting to see future works on more critical population. 

Response 3. Thank you for your consideration of this issue. Due to the composition of IMN and standard ONS products, we decided to reduce the risks to patients as much as possible, therefore we decided to exclude patients at high risk of preoperative hyperglycemia and fluid overload, as these adverse effects could have a negative impact on postoperative outcomes. In addition, we decided to exclude patients with conditions (e.g., malabsorption) that could reduce treatment efficacy. We agree that it would be interesting to include patients with more critical conditions in the future.

Comment 4. In the manuscript the author stated IVF would be given if patients not able to start early oral intake. Pure fluid have no calories nor nutrients, not sure if this would impact the final results. The solution for patients who might not be able to tolerate early oral nutrition s/p surgery not optimal. Would like to have a protocol attached as a supplement material

Response 4. We will track the date of oral refeeding and TPN use to adjust for these potential confounders in the outcome analyses. We agree that a more detailed protocol could better clarify postoperative nutritional interventions, so we have uploaded the algorithm for oral refeeding and/or total parenteral nutrition after radical cystectomy in the Supplementary material.